# Exploring the multilevel determinants of low birth weight in Bangladesh: Understanding implications for targeted public health interventions

**Akher Ali, Md. Kamrul Hasan**\*, **Ruhul Amin**[ID]\*, **Mohammad Alamgir Kabir, Sheikh Mohammad Junaid**[ID]

Department of Statistics and Data Science, Jahangirnagar University, Savar, Dhaka, Bangladesh

\* ruhulaminstat1996@gmail.com (RA); mdkamrulhasan992@gmail.com (KH)

## Abstract

Low birth weight (LBW), a key indicator of impaired fetal growth, is strongly associated with infant morbidity, mortality, and long-term health problems, particularly in developing countries such as Bangladesh. LBW reflects complex public health challenges shaped by interlinked factors across health, education, and socioeconomic sectors, where low maternal education, limited household income, poor nutrition, inadequate healthcare access, and insufficient antenatal care contribute to adverse birth outcomes. Understanding these multilevel determinants is essential for designing effective public health interventions. This study examined the prevalence and determinants of LBW in Bangladesh while accounting for both individual- and community-level factors using data from the 2022 Bangladesh Demographic and Health Survey. The analysis included 5,342 newborns with recorded birth weights from births occurring within the five years preceding the survey. Birth weight was classified as LBW, defined as less than 2,500 grams, or normal birth weight of at least 2,500 grams. Binary logistic regression was applied to identify associated risk factors, and adjusted odds ratios with 95% confidence intervals were reported using a significance level of $p < 0.05$. Multilevel logistic regression models were additionally employed to capture community-level variation and improve model fit and predictive performance. The prevalence of LBW was 9.15%, with significant associations observed for administrative division, place of residence, household wealth index, mode of delivery, number of antenatal care visits, and place of delivery. Random-effects estimates showed reduced cluster-level variability across models, with the intraclass correlation coefficient declining from 10.65% to 6.66% and the median odds ratio decreasing from 1.81 to 1.58, indicating improved explanatory power after adjustment. Overall, LBW in Bangladesh is influenced by maternal characteristics, socioeconomic conditions, healthcare utilization, and contextual factors, underscoring the need for targeted interventions that strengthen maternal healthcare services,

**Data availability statement:** This study was based on an analysis of existing public domain survey datasets that are freely available online with all identifier information removed. The survey was approved by the Ethics Committee in Bangladesh. The authors were granted permission to use the data for independent research purposes. The link of the dataset is https://dhsprogram.com/data/dataset/Bangladesh_Standard-DHS_2022.cfm?flag=0.

**Funding:** The author(s) received no specific funding for this work.

**Competing interests:** The authors have declared that no competing interests exist.

expand antenatal care coverage, and address socioeconomic disparities to reduce LBW prevalence and related risks.

## Introduction

Low birth weight (LBW), as defined by the World Health Organization (WHO) as a birth weight under 2.5 kilograms, remains a critical public health issue worldwide [1]. LBW is a multifactorial condition influenced by a variety of prenatal and postnatal factors. Infants born with LBW face a mortality risk approximately 20 times higher than those with a birth weight exceeding 2,500 grams [2]. It is closely associated with preterm birth, occurring before 37 weeks of gestation, and intrauterine growth restriction (IUGR), both of which contribute to adverse neonatal outcomes [3]. LBW infants are at an increased risk of infections, childhood diseases, and lower survival rates. Long-term effects may include physical and cognitive disabilities, which can influence behavior, learning abilities, and psychosocial development [4]. Notably, LBW is responsible for 40% of all deaths in children under five, with 75% of these fatalities occurring within the first week of life and 25–45% within the first 24 hours [5]. Low birth weight is a critical risk factor for perinatal survival, as well as infant and child mortality, and morbidity during infancy and early childhood (ages 3–8). It plays a significant role in the overall burden of infant mortality [6]. Infants with LBW are more likely to experience digestive and respiratory issues, along with difficulties in feeding, gaining weight, and combating infections, in comparison to infants with normal birth weight [7]. As LBW infants mature into adulthood, they may face cognitive impairments, developmental disorders, physical disabilities, chronic fatigue, depression, and other psychiatric conditions. Additionally, they are at an elevated risk of non-communicable diseases such as hypertension, diabetes, sleep-disordered breathing, and cardiovascular disease [8,9,10]. It is a major factor in neonatal morbidity and mortality, with profound long-term health implications.

The World Health Organization (WHO) and UNICEF estimate that 15–20% of live births globally each year are LBW infants [11]. Moreover, approximately 95% of the 20 million LBW births occur in countries with low to middle socioeconomic development [12]. It is a major contributor to infant mortality in developing regions. Globally, around 15% of all births are classified as LBW, with developing countries experiencing a significantly higher burden [4]. LBW has both immediate and long-term effects, including increased susceptibility to infections, malnutrition, cognitive impairments, and disabilities during childhood and beyond. Additionally, LBW is associated with a higher risk of developing non-communicable diseases such as diabetes and cardiovascular disease in adulthood [13,14]. Each year, LBW is a leading factor in approximately 60%–80% of neonatal fatalities globally [15]. The incidence of low birth weight differs considerably across regions, being most prevalent in low- and middle-income countries, especially among vulnerable populations. Research shows that around 15% of newborns globally are born with LBW, with more than half of these cases occurring in Asia [16]. In East Asia and the Pacific, around 6% of newborns are born with low birth weight, while in South Asia, the prevalence can rise to as high as 28%

[1]. In Bangladesh, the LBW rate has remained relatively stable over recent decades, with rates of 17.7% in 2011, 20% in 2014, 16% in 2017, and 13.5% in 2024. Despite a decline in LBW rates between 2011, 2017, and 2024 Bangladesh's rate remains higher than that of many other developing countries [17].

Efforts to reduce the prevalence of LBW have been a key focus of global health policies, underscoring the need to identify its most significant contributing factors. Numerous studies have examined the risk factors associated with LBW using Demographic and Health Surveys (DHS) across developing countries, including Bangladesh [18,19,20]. These studies primarily assessed socioeconomic and demographic determinants; however, the influence of maternal and child health-related factors on LBW remains largely unexplored [21,22]. Key socioeconomic determinants, such as wealth index, education, family income, occupation, and family size, have been identified as significant predictors of LBW [23,24]. Pregnant women from low socioeconomic backgrounds often face barriers to accessing healthcare services and experience higher levels of food and nutritional insecurity compared to those from wealthier households, increasing their risk of delivering LBW infants. Addressing this issue requires a multifaceted approach that includes improving maternal nutrition, managing pregnancy-related health conditions, and strengthening maternal and perinatal healthcare services [25,26]. This study aims to bridge the existing research gap by exploring the impact of maternal and child health-related factors on LBW.

In Bangladesh, the prevalence of LBW remains high at 13.5%, exceeding that of many other developing nations. This persistent issue raises significant concerns for future health policy development. Therefore, in-depth research is essential to enhance child health outcomes and achieve the health-related Sustainable Development Goals (SDGs) by 2030. The primary objective of this study is to analyze the impact of various contributing factors on LBW using statistical modeling and logistic regression techniques. Specifically, the study aims to assess the prevalence and risk factors associated with LBW in Bangladesh. It seeks to investigate socioeconomic disparities in LBW prevalence and identify key determinants using data from the most recent BDHS 2021–2022. Findings from this study can inform targeted interventions to reduce LBW-related morbidity and mortality, ultimately improving maternal and neonatal health. Moreover, this research aligns with global efforts to lower LBW prevalence, enhance neonatal well-being, and promote healthier future generations in Bangladesh and beyond.

## Materials and methods

### Ethics statement

This study is based on secondary data obtained from the BDHS 2022, a publicly available dataset. The original survey received ethical approval from the Institutional Review Board (IRB) of ICF Macro in Calverton, USA, and the National Ethics Committee of Bangladesh. Informed consent was obtained from all participants during data collection, and all ethical standards were strictly followed. The dataset used in this analysis contains no personally identifiable information, ensuring full confidentiality and privacy of respondents. Therefore, no additional ethical approval was required for this secondary data analysis.

### Sampling method and sample size

This study used data from the BDHS 2022, a nationally representative survey employing a stratified two-stage cluster sampling design. LBW information was obtained from health cards or, when unavailable, from maternal recall, which may introduce some recall bias or measurement error. In the first stage, 675 enumeration areas (237 urban and 438 rural) were selected using probability proportional to size.

A household listing within each EA created the sampling frame, and in the second stage, 45 households per EA were systematically selected. The survey included approximately 61,488 ever-married women aged 15–49, providing detailed birth histories. Sample weights were applied to ensure national, urban-rural, and divisional representativeness. After excluding missing and outlier data, the final analytical dataset comprised 5,342 observations in Fig 1.

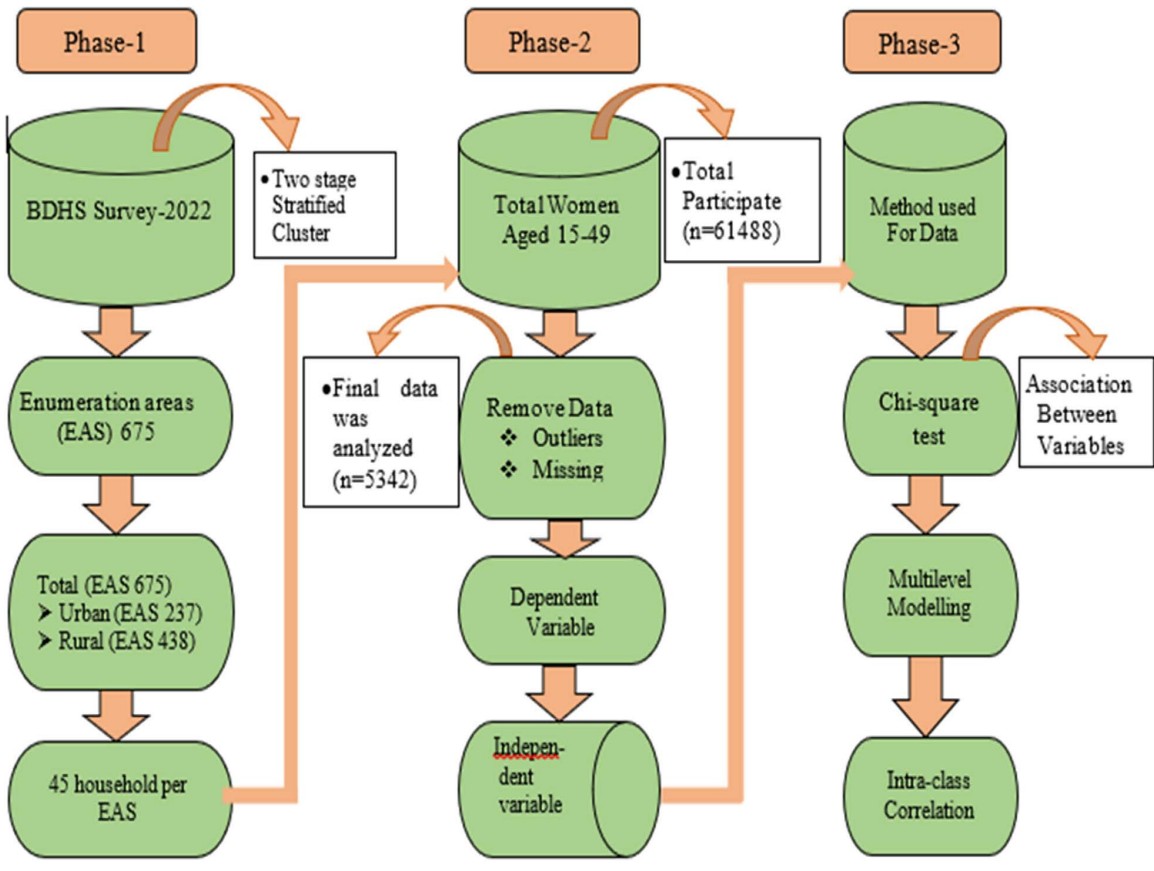

**Fig 1. Design of study.**

## Data collection and outcome variable

Birth weight data were obtained primarily from health cards. When health cards were unavailable, maternal recall was used. While this approach ensures broader coverage, it may introduce recall bias or measurement error, which could affect the accuracy of reported birth weights. The primary outcome variable in this study was birth weight, measured in grams. LBW was classified as less than 2,500 grams, while NBW was defined as 2,500 grams or more. For analysis, the outcome variable was coded as "1" for LBW and "0" for NBW.

## Independent variables

In the current study, we considered several predictors based on the relevant previous works [27–29]. These included mothers age, Division, Education Level, Wealth Index, Residence Type, currently working, sex of the child, child is alive, caesarian delivery, parity, during pregnancy given bought iron tablets/syrups, twin child, Birth interval, decide childcare, ANC visit, Size of child birth, Delivery Place, BMI. A detailed description of predictor types and categorizations is provided in S1 Table.

## Statistical analysis

Data were taken into Stata software then cleaned and weighed by using "svy" function. After excluding missing value and outlier, the final analytical dataset comprised 5,342 observations. Data for nominal and ordinal variables were expressed as a percentage (%), whereas data for

Continuous variables were made in the categorical form. We employed a chi-square test for nominal variables to examine the association between different factors and LBW. A p-value<0.05 determines the statistical significance. The significant predictors were used in the LR model to determine the prominent risk factors of LBW. To account for both individual- and community-level variations, multilevel logistic regression (MLM) was employed, enabling estimation of intra-class correlation (ICC) and random effects. All analyses were performed using Stata version 14.2 and ROC curve in R programming language.

## Model selection procedure

Chi-square tests to assess the degree of interaction among LBW and a number of sociodemographic, reproductive, and healthcare-related variables were first used to find possible predictors of LBW. The variables examined included age, division, education level, wealth index, residence type, working status, sex of the child, child survival status, caesarean delivery, parity, iron tablet or syrup intake during pregnancy, twin child status, birth interval, childcare decision, ANC visits, size of the child at birth, delivery place, and maternal BMI. We applied a backward stepwise elimination approach based on the likelihood ratio test (LRT) to identify the best-fitting logistic regression model. Variables that were significant in the bivariate analysis (p<0.05) were initially included, and non-significant variables were sequentially removed until only those with p<0.05 remained in the final model.

## Logistic regression

Logistic regression, also referred to as the logistic model or logit model, is a widely used approach within generalized linear modeling. It is one of the most fundamental and well-established supervised machine learning (ML) classification techniques. This method is extensively applied in both regression and classification tasks, including the prediction of health conditions such as hypertension, LBW, diabetes, and so on [30–32]. Logistic regression models binary outcome variables by utilizing a set of predictor variables, which can be either discrete or continuous.

Consider $y$ as the outcome variable, representing the binary classification of low birth weight (LBW) or normal birth weight (NBW). Let $X = (x_1, x_2, x_3, ..., x_{11})$ denote the set of predictor variables. The logistic regression model is mathematically expressed as follows:

$$\text{logit}(\pi) = \ln\left(\frac{\pi}{1-\pi}\right) = \alpha + \beta_1 x_1 + \beta_2 x_2 + \beta_3 x_3 + \ldots + \beta_{11} x_{11} \tag{1}$$

Equivalently, by applying exponentiation to both sides of the equation.

$$\frac{\pi}{1-\pi} = e^{\alpha + \beta_1 x_1 + \beta_2 x_2 + \beta_3 x_3 + \ldots + \beta_{11} x_{11}} \tag{2}$$

Here, $\pi$ represents the probability of the event LBW occurring, while $1-\pi$ denotes the probability of the event NBW occurring. The model parameters, $\alpha, \beta_1, \beta_2, \beta_3, ..., \beta_{11}$ are estimated using the maximum likelihood estimation (MLE) method to ensure optimal parameter values for the logistic regression model.

## Multilevel Modeling (MLM)

Multilevel modeling (MLM), also known as hierarchical linear modeling (HLM) or mixed-effects modeling, is a statistical technique designed to analyze nested data structures, accounting for both fixed and random effects. It is particularly useful for modeling intra-class correlation (ICC) and capturing group-level variability [33,34].

A two-level multilevel model is mathematically expressed as follows:

$$Y_{ij} = \beta_0 + \sum_{k=1}^{11} \beta_k x_{k,ij} + \mu_j + \in_{ij} \tag{3}$$

Where, $Y_{ij}$ is the outcome, $\beta_0$, $\beta_k$ are fixed effects, $\mu_j$ represents the group-level random effect, and $\in_{ij}$ is the individual-level residual error.

The ICC quantifies the proportion of total variance explained by group-level differences:

$$ICC = \frac{\sigma_\mu{}^2}{\sigma_\mu{}^2 + \sigma_\in{}^2}$$

(4)

Where, $\sigma_\mu{}^2$ is between-group variance, and $\sigma_\in{}^2$ is within-group variance. ICC was calculated to quantify the proportion of total variance explained by cluster-level differences, highlighting the importance of group-level effects.

## Results

### Sociodemographic and health characteristics

Table 1 presents the distribution of LBW across key sociodemographic and maternal health characteristics among 5,342 participants. The overall prevalence of LBW was 9.15%. Significant regional differences were observed (p<0.01), with LBW ranging from 5.71% in Mymensingh to 13.28% in Dhaka. Maternal age was not significantly associated with LBW (p=0.548). Education showed a small but significant association (p<0.05), where women with higher education exhibited a slightly higher proportion of LBW (12.03%). Household wealth had a significant gradient (p<0.01): LBW increased steadily from 7.57% among the poorest to 12.97% among the richest. Urban mothers had higher LBW prevalence compared to rural mothers (11.58% vs. 8.74%; p<0.05). Maternal employment status and sex of the child were not associated with LBW. Several maternal and child health factors were significantly related to LBW. Children who were not alive at the time of survey had markedly higher LBW prevalence (20.54%; p<0.01). Caesarean deliveries were associated with higher LBW (12.17%) compared with vaginal births (7.36%; p<0.01). Iron supplementation during pregnancy, ANC visits, and wealth appeared counterintuitive, with higher LBW percentages among mothers who received iron supplements or more ANC visits; these patterns likely reflect underlying high-risk pregnancies rather than protective effects. Multiple births showed the strongest association with LBW (p<0.01): more than 30% of twins were LBW compared to 9.04% of singletons. Very short birth intervals (<18 months) were associated with higher LBW (16.58%; p=0501). Decision-making autonomy, place of delivery, parity, and maternal BMI were not significantly associated with LBW. Perceived size of the child at birth showed a strong graded relationship (p<0.01). LBW prevalence was extremely high among infants perceived as "very small" (48.99%) and "smaller than average" (38.51%), but very low among "larger than average" infants (2.72%).

### Adjusted Associations with Low Birth Weight (AOR)

Subsequent logistic regression modeling incorporated all variables that demonstrated statistical significance in the preliminary chi-square analyses in Table 2. The logistic regression analysis reveals several significant associations between LBW and maternal, socioeconomic, and birth-related factors. The multivariable logistic regression model identified several factors independently associated with LBW. After full adjustment, notable geographic variation remained. Compared with Dhaka, mothers residing in Mymensingh (AOR=0.53; 95% CI: 0.34–0.81; p<0.01) and Rajshahi (AOR=0.54; 95% CI: 0.33–0.86; p<0.01) had significantly lower odds of LBW, whereas other divisions did not differ meaningfully (all p>0.05). Residence type showed no independent association, with rural mothers exhibiting comparable odds to urban mothers (AOR=1.10; p=0.538). Similarly, education level and wealth index categories demonstrated no statistically significant effects (all p>0.10), suggesting that socioeconomic factors may act indirectly or be mediated through other covariates. Multiple births had markedly elevated LBW risk: first-born twins (AOR=3.56; 95% CI: 1.59–8.27; p<0.01) and second-born twins (AOR=3.67; 95% CI: 1.62–8.30; p<0.01) were more than three times as likely to be LBW compared with singletons. Caesarean delivery was significantly associated with increased LBW odds (AOR=1.73; 95% CI: 1.26–2.37; p<0.01). Child alive status approached significance, with surviving children showing lower LBW odds (AOR=0.51;

**PLOS Global Public Health**

**Table 1. Frequency distribution of participant by sociodemographic and health experiences of Low Birth Weight (LBW).**

| Variables | Category | n (%) | Low Birth Weight | | P-value |
| --- | --- | --- | --- | --- | --- |
| | | | **Normal** | **Low** | |
| Total Participants | | 5342 | 4853(90.85) | 489(9.15) | |
| Division | Barishal | 570 (10.67) | 512(90.47) | 58(9.53) | <0.01*** |
| | Chittagong | 932 (17.45) | 836(90.07) | 96(9.93) | |
| | Dhaka | 796 (14.90) | 694(86.72) | 102(13.28) | |
| | Khulna | 593 (11.10) | 544(92.14) | 49(7.87) | |
| | Mymensingh | 664 (12.43) | 623(94.29) | 41(5.71) | |
| | Rajshahi | 527 (9.87) | 494(93.5) | 33(6.50) | |
| | Rangpur | 617 (11.55) | 570(92.23) | 47(7.77) | |
| | Sylhet | 643 (12.04) | 580(90.88) | 63(9.12) | |
| Age group | 15-19 | 705 (13.20) | 633(89.27) | 72(10.73) | 0.548 |
| | 20-24 | 1735 (32.48) | 1584(91.34) | 151(8.66) | |
| | 25-29 | 1530 (28.64) | 1390(90.54) | 140(9.46) | |
| | 30-34 | 923 (17.28) | 844(90.57) | 79(9.43) | |
| | 35-39 | 372 (6.96) | 331(88.11) | 41(11.89) | |
| | 40-44 | 69 (1.29) | 64(93.79) | 5(6.21) | |
| | 45-49 | 8 (0.15) | 7(89.41) | 1(10.59) | |
| Education level | No Education | 283 (5.30) | 260(90.61) | 23(9.39) | <0.05** |
| | Primary | 1250 (23.40) | 1154(92.61) | 96(7.39) | |
| | Secondary | 2808 (52.56) | 2543(90.43) | 265 (9.57) | |
| | Higher | 1001 (18.74) | 896(87.97) | 105 (12.03) | |
| Wealth index | Poorest | 1127 (21.10) | 1045(92.43) | 82 (7.57) | <0.01*** |
| | Poorer | 1078 (20.18) | 994(91.96) | 84 (8.04) | |
| | Middle | 1062 (19.88) | 962(90.74) | 100 (9.26) | |
| | Richer | 1049 (19.64) | 942(89.85) | 107 (10.15) | |
| | Richest | 1026 (19.21) | 910(87.03) | 116(12.97) | |
| Residence Type | Urban | 1757 (32.89) | 1573(88.42) | 184(11.58) | <0.05** |
| | Rural | 3585 (67.11) | 3280(91.26) | 305(8.74) | |
| Currently working | No | 4209 (78.79) | 3805(90.18) | 404(9.82) | 0.226 |
| | Yes | 1133 (21.21) | 1048(91.64) | 85(8.34) | |
| Sex of the child | Male | 2738 (51.25) | 2491(90.63) | 247(9.37) | 0.767 |
| | Female | 2604 (48.75) | 2362(90.37) | 242(9.63) | |
| Child is alive | No | 158 (2.96) | 128(79.46) | 30(20.54) | <0.01*** |
| | Yes | 5184 (97.04) | 4725(90.81) | 459(9.19 | |
| Caesarian delivery | No | 2957 (55.35) | 2749(92.64) | 208(7.36) | <0.01*** |
| | Yes | 2374 (44.45) | 2095(87.83) | 279(12.17) | |
| Parity | Parity <5 | 5158 (96.56) | 4684(90.47) | 474(9.53) | 0.7263 |
| | Parity >=5 | 184 (3.44) | 169(91.37) | 15(8.63) | |
| During pregnancy taking iron tablets/syrups | No | 1050 (19.66) | 983(93.65) | 67(6.35) | <0.01*** |
| | Yes | 4013 (75.12) | 3632(89.99) | 381(10.01) | |
| | Don't know | 1 (0.02) | 1(100) | 0(0) | |
| Twin child | Single birth | 5232 (97.94) | 4778(90.96) | 454(9.04) | <0.01*** |
| | 1st of multiple | 55 (1.03) | 38(68) | 17(32) | |
| | 2nd of multiple | 55 (1.03) | 37(65.44) | 18(34.56) | |

*(Continued)*

**Table 1.** (Continued)

| Variables | Category | n (%) | Low Birth Weight | | P-value |
|---|---|---|---|---|---|
| | | | Normal | Low | |
| Birth interval | Very Short<(18m) | 128 (2.40) | 112(83.42) | 16(16.58) | 0.0501 |
| | Short (18-23m) | 194 (3.63) | 182(93.98) | 12(6.02) | |
| | Optimal (24-59m) | 1355 (25.37) | 1226(90.40) | 129(9.60) | |
| | Long (≥60m) | 3665 (68.61) | 3333(90.60) | 332(9.40) | |
| Decide healthcare | Respondent alone | 444 (8.31) | 401(89.01) | 43(10.99) | 0.7658 |
| | Response & husband | 3350 (62.71) | 3040(90.58) | 310(9.42) | |
| | Husband alone | 1315 (24.62) | 1196(90.40) | 119(9.60) | |
| | Someone else | 172 (3.22) | 157(91.19) | 15(8.81) | |
| | Other | 12 (0.22) | 12(100) | 0(0) | |
| Size of child birth | Very large | 32 (0.60) | 29(91.56) | 3(8.44) | <0.01*** |
| | Larger than average | 488 (91.14) | 477(97.28) | 11(2.72) | |
| | Average | 4091 (76.58) | 3897(95.15) | 194(4.85) | |
| | Smaller than average | 677 (12.67) | 415(61.49) | 262(38.51) | |
| | Very small | 44 (0.82) | 25(51.01) | 19(48.99) | |
| | Don't know | 10 (0.19) | 10(100) | 0(0) | |
| ANC visits | <=2 Visit | 2088 (39.09) | 1954(93.28) | 134(6.72) | <0.01*** |
| | >2 Visit | 3254 (60.91) | 2899(88.66) | 355(11.34) | |
| Delivery Place | Home | 183 (3.43) | 148(80.46) | 35(19.54) | 0.1032 |
| | Hospital | 2382 (44.59) | 2084(87.02) | 298(12.98) | |
| | Others | 17 (0.32) | 14(81.16) | 3(18.84) | |
| BMI | Under weight | 401 (7.51) | 363(90.02) | 38(9.98) | 0.3602 |
| | Normal weight | 1529 (28.62) | 1395(90.68) | 134(9.32) | |
| | Over weight | 575 (10.76) | 519(88.27) | 56(11.73) | |
| | Obese | 2816 (52.71) | 2560(91.01) | 256(8.99) | |

**Note:** p-value<0.01, '***', p<0.05, '**'

p = 0.069), consistent with the documented link between LBW and neonatal mortality. Perceived size at birth was a strong predictor, showing a powerful graded association. Compared with infants perceived as "very large," those assessed as "smaller than average" had substantially higher LBW odds (AOR = 6.13; p < 0.01) and those perceived as "very small" had the highest risk (AOR = 13.85; p < 0.01). These findings confirm the reliability of maternal perception in reflecting true birth outcomes. During Pregnancy taking iron tablet/syrups showed marginal significance (AOR = 1.36; p = 0.71), likely indicating reverse causation, where high-risk pregnancies receive more supplements. Reproductive factors also contributed meaningfully. A short birth interval of 18–23 months significantly reduced LBW odds compared with very short intervals (<18 months) (AOR = 0.32; p < 0.05). Optimal (AOR = 0.51; p = 0.075) and long intervals (AOR = 0.48; p = 0.055) demonstrated borderline protective effects. Mothers with >2 ANC visits showed significantly higher odds of LBW (AOR = 1.48; p < 0.01), likely reflecting increased care-seeking among pregnancies already at elevated risk.

### Unadjusted Associations with Low Birth Weight (UOR)

The unadjusted logistic regression analysis identified several factors that were significantly associated with LBW were presented in S2 Table. Substantial geographic variation was observed across administrative divisions. Compared with Dhaka, the odds of LBW were significantly lower in Khulna (OR = 0.56; p < 0.01), Mymensingh (OR = 0.40; p < 0.01),

**Table 2. Adjusted odds ratio (AOR) from Logistic regression model for LBW.**

| Factors | Category | Adjusted OR (95% C.I.) | p-value |
|---|---|---|---|
| Division | Dhaka | Reference | |
| | Barishal | 0.95 (0.60,1.49) | 0.812 |
| | Chattogram | 0.88 (0.59,1.30) | 0.521 |
| | Khulna | 0.66 (0.41,1.06) | 0.087 |
| | Mymensingh | 0.53 (0.34,0.81) | <0.01*** |
| | Rajshahi | 0.54 (0.33,0.86) | <0.01*** |
| | Rangpur | 0.73 (0.44,1.23) | 0.233 |
| | Sylhet | 0.72 (0.44,1.19) | 0.199 |
| Residence type | Urban | Reference | |
| | Rural | 1.1 (0.81,1.49) | 0.538 |
| Education Level | No education | Reference | |
| | Primary | 0.81 (0.46,1.41) | 0.453 |
| | Secondary | 0.90 (0.54,1.48) | 0.671 |
| | Higher | 1.04 (0.60,1.81) | 0.889 |
| Wealth Index | Poorest | Reference | |
| | Poorer | 1.10 (0.68,1.78) | 0.689 |
| | Middle | 1.11 (0.71,1.74) | 0.638 |
| | Richer | 1.40 (0.90,2.19) | 0.139 |
| | Richest | 1.40 (0.87,2.24) | 0.165 |
| Twins Child | Single birth | Reference | |
| | $1^{st}$ of multiple | 3.56 (1.59,8.27) | <0.01*** |
| | $2^{nd}$ of multiple | 3.67 (1.62,8.30) | <0.01*** |
| Child is alive | No | Reference | |
| | Yes | 0.51 (0.25, 1.05) | 0.069 |
| Caesarian delivery | No | Reference | |
| | Yes | 1.73 (1.26, 2.37) | <0.01*** |
| Size of child birth | Very large | Reference | |
| | Larger than average | 0.28 (0.06, 1.40) | 0.121 |
| | Average | 0.48 (0.12, 1.97) | 0.307 |
| | Smaller than average | 6.13 (1.51, 24.98) | <0.01*** |
| | Very small | 13.85 (2.79, 68.84) | <0.01*** |
| | Don't know | 1 | |
| During Pregnancy taking iron tablet/syrups | No | Reference | |
| | Yes | 1.36 (0.97, 1.90) | 0.071 |
| | Don't know | 1 | |
| Birth interval | Very Short<(18m) | Reference | |
| | Short (18-23m) | 0.32 (0.12, 0.89) | <0.05** |
| | Optimal (24-59m) | 0.51 (0.24, 1.07) | 0.075 |
| | Long (≥60m) | 0.48 (0.23, 1.01) | 0.055 |
| ANC visits | <=2 Visit | Reference | |
| | >2 Visit | 1.48 (1.09, 2.01) | <0.01*** |

**Note:** p-value<0.01, '***', p<0.05, '**'

Rajshahi (OR = 0.45; p < 0.01), Rangpur (OR = 0.55; p < 0.01), and Sylhet (OR = 0.66; p < 0.05). Barishal and Chattogram showed marginal reductions (both p < 0.10). Rural residence was associated with lower odds of LBW compared with urban areas (OR = 0.73; p < 0.05). Socioeconomic indicators exhibited mixed patterns. Maternal education showed no significant associations (all p > 0.25). Wealth index demonstrated a clear gradient, with mothers in the richest households showing significantly higher odds of LBW (OR = 1.82; p < 0.01), while other wealth categories did not differ significantly from the poorest group. Biological and obstetric factors displayed some of the strongest associations. Multiple births were a major predictor: the first-born (OR = 4.73; p < 0.01) and second-born twins (OR = 5.31; p < 0.01) exhibited more than a four- to fivefold increase in LBW odds compared to singletons. Child survival status was strongly associated, with surviving children showing substantially lower odds of LBW (OR = 0.39; p < 0.01). Caesarean delivery was also linked to higher LBW odds (OR = 1.74; p < 0.01). Perceived size at birth demonstrated a pronounced gradient. Infants perceived as "smaller than average" (OR = 6.79; p < 0.01) and "very small" (OR = 10.42; p < 0.01) had markedly increased LBW odds relative to "very large" infants, indicating strong alignment between maternal perception and actual birth weight. Several maternal health and reproductive characteristics were also significant. Iron supplementation during pregnancy showed higher LBW odds (OR = 1.64; p < 0.01), likely reflecting care-seeking patterns among high-risk pregnancies. Birth interval demonstrated a protective effect: short (18–23 months), optimal (24–59 months), and long (≥60 months) intervals significantly reduced LBW odds (all p < 0.05) compared with very short intervals (<18 months). Mothers with more than two ANC visits had significantly elevated LBW odds (OR = 1.78; p < 0.01), again consistent with higher utilization among pregnancies experiencing complications.

## Multivariate multilevel model

Table 3 presents the multivariate multilevel logistic regression assessing the determinants of LBW in Bangladesh across four sequential models. Model 0 (null model) included no explanatory variables and estimated the baseline community-level variance. The model yielded a variance of 0.39 (SE = 0.11) and an intraclass correlation coefficient (ICC) of 10.65%, indicating that a substantial proportion of the variation in LBW was attributable to community-level differences. The median odds ratio (MOR) was 1.81, suggesting moderate heterogeneity across clusters. The AIC was 3255.92, serving as the baseline model fit. Model 1 introduced individual-level factors. Mother's education showed mixed associations: primary (aOR = 0.95, 95% CI: 0.54–1.68; p = 0.871), secondary (aOR = 1.20, 95% CI: 0.70–2.06; p = 0.513), and higher education (aOR = 1.34, 95% CI: 0.75–2.41; p = 0.323) did not significantly affect LBW compared with no education. Birth interval was also nonsignificant, with short (18–23 months) (aOR = 0.47, 95% CI: 0.19–1.18; p = 0.107), optimal (24–59 months) (aOR = 0.68, 95% CI: 0.35–1.34; p = 0.267), and long (≥60 months) (aOR = 0.63, 95% CI: 0.33–1.20; p = 0.162) intervals showing no significant association with LBW. ANC visits >2 was significantly protective against LBW (aOR = 1.87, 95% CI: 1.44–2.44; p < 0.01). Being a twin birth strongly increased LBW risk: first of multiple births (aOR = 3.25, 95% CI: 1.94–7.56; p < 0.01) and second of multiple births (aOR = 4.37, 95% CI: 2.16–8.84; p < 0.01) compared with singletons. Child alive status was not significant (aOR = 0.60, 95% CI: 0.32–1.13; p = 0.114). Size of child birth was a strong predictor: smaller than average (aOR = 5.38, 95% CI: 1.55–18.68; p < 0.01) and very small (aOR = 8.72, 95% CI: 2.13–35.78; p < 0.01) significantly increased LBW odds compared with very large size, while larger than average (aOR = 0.18, 95% CI: 0.05–0.71; p < 0.05) was protective. Iron tablets/syrups during pregnancy showed borderline effects: yes (aOR = 1.46, 95% CI: 1.06–2.02; p < 0.05) and don't know (aOR = 1.78, 95% CI: 1.14–2.69; p = 0.068). Community-level variance decreased slightly to 0.34 (SE = 0.14), with a proportional change in variance (PCV) of 12.82%, ICC of 9.28%, and MOR of 1.74. The mean VIF was 6.81, indicating no multicollinearity, and the AIC decreased to 2424.30, showing modest improvement in model fit. Model 2 included only community-level variables. Division and household wealth showed significant associations with LBW. Children from Mymensingh (aOR = 0.60, 95% CI: 0.38–0.96; p < 0.05) and Rajshahi (aOR = 0.55, 95% CI: 0.34–0.90; p < 0.05) had significantly lower odds of LBW compared with Barishal, while other divisions were nonsignificant. Household wealth exhibited mixed effects: middle-income households (aOR = 1.11, 95% CI: 0.80–1.54;

PLOS Global Public Health

**Table 3. Multivariate multilevel model assessing the effects of individual and community level factors on LBW.**

| Factors | Model_0 | Model_1 | | Model_2 | | Model_3 | |
|---|---|---|---|---|---|---|---|
| | | AOR (95% C.I.) | p-value | AOR (95% C.I.) | p-value | AOR (95%C.I.) | p-value |
| **Measures of association (fixed-effects)** | | | | | | | |
| **Mother's Education level** | | | | | | | |
| No education | | Reference | | | | | |
| Primary | | 0.95 (0.54, 1.68) | 0.871 | | | 0.95 (0.54, 1.68) | 0.871 |
| Secondary | | 1.20 (0.70, 2.06) | 0.513 | | | 1.20 (0.70, 2.06) | 0.513 |
| Higher | | 1.34 (0.75, 2.41) | 0.323 | | | 1.34 (0.75, 2.41) | 0.323 |
| **Birth interval** | | | | | | | |
| Short<(18m) | | Reference | | | | | |
| Short (18-23m) | | 0.47 (0.19, 1.18) | 0.107 | | | 0.47 (0.19, 1.18) | 0.106 |
| Optimal(24-59m) | | 0.68 (0.35, 1.34) | 0.267 | | | 0.66 (0.33, 1.31) | 0.237 |
| Long (≥60m) | | 0.63 (0.33, 1.20) | 0.162 | | | 0.61 (0.32, 1.19) | 0.147 |
| **ANC visits** | | | | | | | |
| ≤2 Visits | | Reference | | | | | |
| >2 Visits | | 1.87 (1.44, 2.44) | <0.01 | | | 1.54 (1.17, 2.02) | <0.01 |
| **Child is twin** | | | | | | | |
| Single birth | | Reference | | | | | |
| 1st of multiple | | 3.25 (1.94, 7.56) | <0.01 | | | 3.78 (1.74, 7.56) | <0.01 |
| 2nd of multiple | | 4.37 (2.16, 8.84) | <0.01 | | | 3.82 (1.84, 7.91) | <0.01 |
| **Child is alive** | | | | | | | |
| No | | Reference | | | | | |
| Yes | | 0.60 (0.32, 1.13) | 0.114 | | | 0.59 (0.31, 1.13) | 0.112 |
| **Size of Child Birth** | | | | | | | |
| Very large | | Reference | | | | | |
| Larger than average | | 0.18 (0.05, 0.73) | <0.05 | | | 0.18 (0.05, 0.72) | <0.05 |
| Average | | 0.37 (0.11, 1.29) | 0.119 | | | 0.37 (0.11, 1.27) | 0.114 |
| Smaller than average | | 5.38 (1.55, 18.68) | <0.01 | | | 5.38 (1.55, 18.70) | <0.01 |
| Very small | | 8.72 (2.13, 35.78) | <0.01 | | | 9.66 (2.35, 39.75) | <0.01 |
| **During pregnancy Taking iron tablets/syrups** | | | | | | | |
| No | | Reference | | | | | |
| Yes | | 1.46 (1.06, 2.02) | <0.05 | | | 1.36 (0.98, 1.88) | 0.067 |
| Don't know | | 1.78 (1.14, 2.69) | 0.068 | | | 1.47 (1.00, 1.95) | <0.05 |
| **Division** | | | | | | | |
| Barishal | | Reference | | | | | |
| Chattogram | | | | 1.08 (0.73, 1.60) | 0.705 | 0.86 (0.55, 1.35) | 0.519 |
| Dhaka | | | | 1.21 (0.82, 1.80) | 0.341 | 1.00 (0.63, 1.57) | 0.988 |
| Khulna | | | | 0.70 (0.45, 1.09) | 0.117 | 0.71 (0.43, 1.16) | 0.169 |
| Mymensingh | | | | 0.60 (0.38, 0.96) | <0.05 | 0.55 (0.33, 0.93) | <0.05 |
| Rajshahi | | | | 0.55 (0.34, 0.90) | <0.05 | 0.53 (0.31, 0.92) | <0.05 |
| Rangpur | | | | 0.75 (0.48, 1.17) | 0.204 | 0.73 (0.44, 1.21) | 0.222 |
| Sylhet | | | | 1.06 (0.69, 1.63) | 0.798 | 0.79 (0.48, 1.30) | 0.357 |
| **Wealth index** | | | | | | | |
| Poorest | | Reference | | | | | |
| Poorer | | | | 0.97 (0.70, 1.35) | 0.850 | 1.12 (0.76, 1.64) | 0.571 |
| Middle | | | | 1.11 (0.80, 1.54) | 0.543 | 1.31 (0.89, 1.94) | 0.172 |

*(Continued)*

**Table 3.** (Continued)

| Factors | Model_0 | Model_1 | | Model_2 | | Model_3 | |
|---|---|---|---|---|---|---|---|
| | | AOR (95% C.I.) | p-value | AOR (95% C.I.) | p-value | AOR (95%C.I.) | p-value |
| Richer | | | | 1.12 (0.80, 1.56) | 0.506 | 1.53 (1.02, 2.30) | <0.05 |
| Richest | | | | 1.08 (0.76, 1.54) | 0.658 | 1.36 (0.87, 2.11) | 0.177 |
| **Residence type** | | | | | | | |
| Urban | | Reference | | | | | |
| Rural | | | | 0.95 (0.75, 1.20) | 0.680 | 1.05 (0.81, 1.38) | 0.702 |
| **Delivery place** | | | | | | | |
| Home | | Reference | | | | | |
| Hospitals | | | | 1.78 (1.44, 2.20) | <0.01 | 1.81 (1.40, 2.33) | <0.01 |
| **Measures of variation (random-effects)** | | | | | | | |
| Variance (SE) | 0.39 (0.11) | 0.34 (0.14) | | 0.26 (0.10) | | 0.23 (0.13) | |
| PCV | Reference | 12.82% | | 33.33% | | 41.03% | |
| ICC | 10.65% | 9.28% | | 7.27% | | 6.66% | |
| MOR | 1.81 | 1.74 | | 1.63 | | 1.58 | |
| **Model evaluation** | | | | | | | |
| Log Likelihood | -1625.96 | -1196.15 | | -1590.41 | | -1170.34 | |
| AIC | 3255.92 | 2424.30 | | 3210.83 | | 2398.69 | |

Note: p-value<0.01, '***', p<0.05, '**'

Ref. = Reference category; AOR = Adjusted Odds Ratio; CI = Confidence Interval.

Model 0 represents the null model (intercept-only), which excludes any predictor variables.

Model 1 incorporates individual-level variables exclusively (mean variance inflation factor [VIF] = 6.81).

Model 2 includes only community-level variables (mean VIF = 1.77).

Model 3 combines both individual- and community-level predictors (mean VIF = 4.42).

p = 0.543) and richer households (aOR = 1.12, 95% CI: 0.80–1.56; p = 0.506) had higher odds of LBW, whereas poorer (aOR = 0.97, 95% CI: 0.70–1.35; p = 0.850) and richest households (aOR = 1.08, 95% CI: 0.76–1.54; p = 0.658) were non-significant. Residence type (rural: aOR = 0.95, 95% CI: 0.75–1.20; p = 0.680) was not significantly associated with LBW. Delivery at hospitals significantly increased LBW odds compared with home delivery (aOR = 1.78, 95% CI: 1.44–2.20; p < 0.01). Community-level variance decreased slightly to 0.26 (SE = 0.10), with a proportional change in variance (PCV) of 33.33%, ICC of 7.27%, and MOR of 1.63, indicating a moderate reduction in between-community variation. The mean VIF was 1.77, confirming no severe multicollinearity, and the AIC decreased to 3210.83, suggesting improved model fit.

Model 3 incorporated both individual- and community-level factors. Among individual-level factors, ANC visits >2 significantly increased the odds of LBW (aOR = 1.54, 95% CI: 1.17–2.02; p < 0.01). Twin births were strongly associated with higher LBW risk, with the first of multiple births (aOR = 3.78, 95% CI: 1.74–7.56; p < 0.01) and the second of multiple births (aOR = 3.82, 95% CI: 1.84–7.91; p < 0.01) showing markedly higher odds compared with single births. Size of child birth was also a significant predictor: larger than average (aOR = 0.18, 95% CI: 0.05–0.72; p < 0.05) and average size (aOR = 0.37, 95% CI: 0.11–1.27; p = 0.114) were associated with lower odds, while smaller than average (aOR = 5.38, 95% CI: 1.55–18.70; p < 0.01) and very small (aOR = 9.66, 95% CI: 2.35–39.75; p < 0.01) children had substantially higher odds of LBW. Taking iron tablets/syrups during pregnancy showed a borderline effect (aOR = 1.36, 95% CI: 0.98–1.88; p = 0.067). Mother's education and child alive status were not significantly associated with LBW. Community-level factors showed mixed effects. Children from Mymensingh (aOR = 0.55, 95% CI: 0.33–0.93; p < 0.05) and Rajshahi (aOR = 0.53, 95% CI: 0.31–0.92; p < 0.05) had lower odds of LBW compared with Barishal, while other divisions were nonsignificant.

Household wealth showed that richer households had higher odds of LBW (aOR = 1.53, 95% CI: 1.02–2.30; $p<0.05$), while other wealth categories were nonsignificant. Residence type (rural: aOR = 1.05, 95% CI: 0.81–1.38; $p=0.702$) was nonsignificant. Hospital delivery remained associated with higher LBW risk (aOR = 1.81, 95% CI: 1.40–2.33; $p<0.01$). Community-level variance decreased further to 0.23 (SE = 0.13), with a proportional change in variance (PCV) of 41.03%, ICC of 6.66%, and MOR of 1.58, indicating that inclusion of both individual- and community-level predictors explained a substantial portion of the between-community variation. The mean VIF was 4.42, suggesting no severe multicollinearity, and the AIC dropped to 2398.69, indicating the best model fit among all models in Table 3.

Fig 2 presents the ROC curves and corresponding area under the curve (AUC) values for four multivariate multilevel mixed-effects models predicting LBW. The null model (Model 0) had an AUC of 0.545, indicating no discriminatory ability. Adding individual-level factors in Model 1 raised the AUC to 0.783, showing modest improvement in predictive accuracy based on socio-demographic characteristics. Model 2, which included only community-level variables, yielded a slightly lower AUC of 0.593, suggesting that household context alone was less effective in distinguishing LBW outcomes. The full model (Model 3) reached the highest AUC of 0.793, providing the best predictive performance and highlighting the value of integrating individual, and community-level determinants.

## Discussion

This study provides a concise summary of the key determinants of LBW identified from a nationally representative sample of Bangladeshi mothers. Our findings show that LBW remains a significant public health concern, with notable geographic disparities, strong biological influences, and several maternal and reproductive factors contributing to elevated risk. Multiple births, perceived small size at birth, caesarean delivery, and very short birth intervals emerged as the strongest predictors of LBW, while certain regions particularly Mymensingh and Rajshahi exhibited consistently lower odds compared with Dhaka. Patterns suggesting reverse causation were also evident, as mothers with more ANC visits and those receiving iron supplementation showed higher LBW odds, likely reflecting increased care-seeking in high-risk pregnancies. Overall, these results highlight that both individual- and community-level determinants continue to shape LBW outcomes in Bangladesh. The prevalence of LBW varies significantly across regions, with the highest rates observed in South Asia

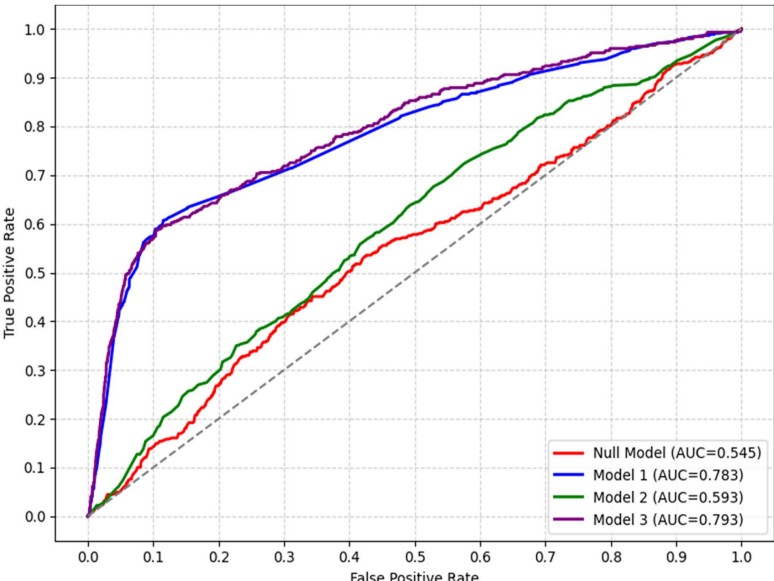

**Fig 2. Model comparison using ROC curve for multivariate multilevel models.**

and sub-Saharan Africa. For example, in South Asia, the prevalence is estimated to be around 26%, while sub-Saharan Africa has a rate of approximately 14% [35]. In contrast, high-income countries report lower rates, with a global average of about 6–8% in Europe and North America [36]. Despite progress in some regions, LBW continues to be a major contributor to neonatal mortality and long-term health challenges, particularly in low- and middle-income countries, where maternal health and healthcare access remain significant barriers. In Bangladesh, the prevalence of low birth weight remains alarmingly high, with estimates ranging from 28% to 32% of all live births, contributing significantly to neonatal morbidity and mortality [37,38].

Our study on LBW in Bangladesh reveals several factors that align with existing research. Regional disparities were evident, with Barishal showing the highest incidence of LBW (9.53%), which is consistent with studies linking regional healthcare access and socioeconomic conditions to birth outcomes [39]. Education level also played a significant role, with women having no education experiencing higher LBW rates (9.39%), supporting the idea that maternal education improves prenatal care and birth outcomes [40]. Additionally, our findings showed a higher LBW prevalence in urban areas (11.58%) compared to rural areas (8.74%), likely due to factors like pollution, restricted access to healthcare, and lifestyle differences in urban settings [41]. We also observed that women with fewer than two antenatal care visits had a higher LBW rate (6.72%), emphasizing the critical role of prenatal care in preventing LBW [40]. The study also confirmed that Caesarian deliveries were more likely to result in LBW (12.17%), supporting the idea that complications during delivery, such as preeclampsia and maternal infections, can lead to adverse birth outcomes [42]. However, factors like maternal age and BMI showed less significant effects in our analysis, in line with some studies suggesting the multifactorial nature of LBW [43]. The null model yielded an ICC of approximately 0.11, indicating moderate community-level clustering in LBW. This declined to about 0.07 in the fully adjusted model, suggesting that the included covariates explained part of the between-community variation, although meaningful contextual differences remained. Notably, our ICC values fall within the range (≈4.9–18.6%) reported in previous LBW studies, further supporting the presence of cluster-level heterogeneity [44,45]. Overall, our findings underscore the need for targeted interventions in maternal education, healthcare access, and addressing regional disparities to reduce LBW rates in Bangladesh.

## Conclusion

In conclusion, this study identifies key factors influencing LBW in Bangladesh, reaffirming insights from prior research. Multiple births, smaller perceived birth size, and regional disparities were the strongest determinants of low birth weight. Although ANC visits and iron intake showed initial associations, their effects diminished after adjustment, suggesting underlying maternal risks. Strengthening maternal care and addressing community-level variations could substantially reduce LBW in Bangladesh. A collaborative effort among healthcare providers, policymakers, and local communities is essential to effectively address the burden of LBW and ensure healthier births across Bangladesh.

## Strength and Limitation

The strengths of this study include its large, nationally representative sample of 5,342 participants from multiple divisions of Bangladesh, providing valuable insights into regional disparities in LBW. The comprehensive analysis of socio-economic, educational, and healthcare-related factors enhances the understanding of LBW determinants, offering critical data for public health interventions. However, this study is limited by its cross-sectional design, which precludes causal inference, and reliance on maternal recall for a portion of birth weights, potentially introducing measurement bias. Key clinical variables, including gestational age, maternal hemoglobin, and detailed dietary intake, were unavailable, leaving residual confounding unaccounted for. Additionally, while nationally representative, some marginalized populations may be underrepresented, and categorization of continuous variables may have reduced precision. Despite these constraints, the findings offer robust, population-level insights into the determinants of low birth weight in Bangladesh.

## Policy implications

The findings of this study have clear implications for maternal and child health policy in Bangladesh. Targeted interventions for high-risk pregnancies, particularly multiple births and infants at risk of being small for gestational age, could reduce the prevalence of LBW. Strengthening antenatal care coverage, improving maternal nutrition, and ensuring timely iron supplementation can mitigate underlying maternal risks. Additionally, addressing regional disparities in healthcare access and quality is essential to promote equitable outcomes. Policymakers can leverage these insights to prioritize resource allocation, design community-level programs, and implement evidence-based strategies aimed at reducing LBW nationwide.

## Recommendation

Based on this study's findings, several key recommendations can be made to reduce LBW in Bangladesh. Firstly, high-risk pregnancies, particularly multiple births, should receive specialized monitoring and care during gestation, as prior research has shown targeted prenatal care reduces LBW risk. Secondly, maternal nutrition should be improved, and early identification of growth-restricted fetuses should be prioritized, focusing on infants at risk of being small for gestational age. Thirdly, regional disparities should be addressed by improving access to quality maternal and neonatal healthcare services in underperforming areas. Finally, antenatal care coverage and iron supplementation programs should be strengthened to ensure that mothers most at risk receive adequate support, consistent with evidence that increased ANC visits and micronutrient supplementation reduce LBW incidence.

## Future Research

While this study provides nationally representative evidence on determinants of LBW in Bangladesh, several avenues for future research remain. Prospective cohort studies are needed to validate causal relationships, particularly for factors such as ANC visits, iron supplementation, and maternal nutrition, which may be influenced by unmeasured confounders or recall bias. Further investigation into biological and environmental determinants, including maternal micronutrient status, intrauterine growth patterns, and regional healthcare accessibility, could provide deeper insights into mechanisms underlying LBW. Additionally, incorporating longitudinal and geospatial analyses would help understand temporal trends and spatial disparities, enabling more targeted interventions to reduce LBW at both individual and community levels.

## Supporting information

**S1 Table. Variables included in the study and their coding for analysis.**
(DOCX)

**S2 Table. Logistic regression model for examining the relationship between Low Birth Weight (LBW) and others factors for unadjusted odds ratio.**
(DOCX)

## Acknowledgments

The authors sincerely acknowledge the Bangladesh Demographic and Health Survey (BDHS-2022) for generously providing the datasets used in this analysis.

## Author contributions

**Conceptualization:** Akher Ali, Md. Kamrul Hasan, Ruhul Amin, Mohammad Alamgir Kabir.
**Data curation:** Akher Ali, Md. Kamrul Hasan, Ruhul Amin.

**Formal analysis:** Akher Ali, Md. Kamrul Hasan, Ruhul Amin, Sheikh Mohammad Junaid.

**Methodology:** Akher Ali, Md. Kamrul Hasan, Ruhul Amin.

**Software:** Md. Kamrul Hasan, Ruhul Amin.

**Supervision:** Mohammad Alamgir Kabir.

**Validation:** Akher Ali.

**Visualization:** Akher Ali, Md. Kamrul Hasan, Ruhul Amin.

**Writing – original draft:** Akher Ali, Md. Kamrul Hasan, Ruhul Amin, Sheikh Mohammad Junaid.

**Writing – review & editing:** Akher Ali, Ruhul Amin, Mohammad Alamgir Kabir.

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
