## [Decision Letter · Decision Letter 0]

22 Oct 2025

PGPH-D-25-00823

Exploring the Multilevel Determinants of Low Birth Weight in Bangladesh: Understanding Implications for Targeted Public Health Interventions

Dear Dr. Amin,

Thank you for submitting your manuscript to PLOS Global Public Health. After careful consideration, we feel that it has merit but does not fully meet PLOS Global Public Health’s publication criteria as it currently stands. Therefore, we invite you to submit a revised version of the manuscript that addresses the points raised during the review process.

Please note that we have only been able to secure a single reviewer to assess your manuscript. We are issuing a decision on your manuscript at this point to prevent further delays in the evaluation of your manuscript. Please be aware that the editor who handles your revised manuscript might find it necessary to invite additional reviewers to assess this work once the revised manuscript is submitted. However, we will aim to proceed on the basis of this single review if possible.

Could you please carefully revise the manuscript to address all comments raised?

A rebuttal letter that responds to each point raised by the editor and reviewer(s). You should upload this letter as a separate file labeled 'Response to Reviewers'.

We look forward to receiving your revised manuscript.

Kind regards,

Helen Howard

Staff Editor

Journal Requirements:

Additional Editor Comments (if provided):

Reviewers' comments:

Reviewer's Responses to Questions

**Comments to the Author**

1. Does this manuscript meet PLOS Global Public Health’s publication criteria?

Reviewer #1: Partly

2. Has the statistical analysis been performed appropriately and rigorously?

Reviewer #1: Yes

3. Have the authors made all data underlying the findings in their manuscript fully available (please refer to the Data Availability Statement at the start of the manuscript PDF file)?

Reviewer #1: Yes

4. Is the manuscript presented in an intelligible fashion and written in standard English?

Reviewer #1: Yes

Reviewer #1: This study uses data from the 2022 Bangladesh Demographic and Health Survey to examine the prevalence and risk factors of low birth weight (LBW). Among 5,342 births, 9.15% were LBW, with significant associations found for maternal age, education, wealth, residence, antenatal care visits, delivery characteristics, and gestational age. The findings underscore LBW as a persistent public health issue requiring targeted maternal health interventions. My comments are below:

General comments

Repetition of text: In a few places (e.g., twin births as a predictor), the manuscript repeats points already made. You may wish to review the results section to streamline these and avoid redundancy.

Specific comments:

Background: Adding some text on how the intersectoral linkage between health, income and education are important will help lay a stronger foundation for the paper to be relevant. It will also support the choice of type of analysis.

Results section:

The results are described in considerable detail. It might be helpful to focus on statistically significant predictors, while mentioning non-significant ones only when the adjusted analysis suggests a change. This would make the results section more concise and focused.

Descriptive table: The paper would benefit from a single, consolidated table describing the characteristics of the study population. This could replace the first table currently presented. It can also concise the results section as the first three tables essentially gives the same message. The unadjusted results can be described briefly with tables removed in annexes. The analysis has the opportunity to highlight on possible discrepancies in LBW across different groups and shed light on existing inequality

Model selection: It would be useful to clarify whether stepwise inclusion methods (forward or backward elimination) were considered when identifying the best-fitting regression model.

Conciseness of results: To reduce repetition, you might consider moving the first two or three detailed tables to the appendix as supplementary material, while presenting the key findings in a more concise form in the main text.

Data collection details: A short description of how LBW data were collected, along with any potential limitations (e.g., recall bias, measurement error), would add valuable context.

Discussion: The opening paragraph of the discussion currently overlaps with the background. It could instead be used to provide a clear summary of the main findings.

Expanding the discussion of how these results compare with or add to existing research could strengthen the manuscript’s contribution. Some of it has already been done but needs more.

Discussion – limitations: Highlighting the key limitations of the analysis (data, methodology, generalisability) would provide a more balanced interpretation of findings.

Policy implications: It may be helpful to draw out the policy relevance of the findings—for example, how they could inform maternal and child health interventions to reduce LBW.

Evidence-based recommendations: Recommendations could be reinforced with references to existing research (e.g., evidence that increased antenatal care visits reduce LBW risk).

Future research: A brief note on what additional research is needed would enhance the discussion.

Acknowledgements: Since DHS is not an organisation but a publicly available dataset managed by partner institutions, a formal acknowledgement may not be necessary.

**Do you want your identity to be public for this peer review?** For information about this choice, including consent withdrawal, please see our Privacy Policy

Reviewer #1: No

---

## [Editor Report · Decision Letter 1]

28 Dec 2025

Exploring the Multilevel Determinants of Low Birth Weight in Bangladesh: Understanding Implications for Targeted Public Health Interventions

PGPH-D-25-00823R1

Dear Mr. Amin,

We are pleased to inform you that your manuscript 'Exploring the Multilevel Determinants of Low Birth Weight in Bangladesh: Understanding Implications for Targeted Public Health Interventions' has been provisionally accepted for publication in PLOS Global Public Health.

Best regards,

Shiyam Sunder, MBBS, MSc epidemiology, PhD

Academic Editor